# In Search of Disentanglement in Tandem Mass Spectrometry Datasets

**DOI:** 10.3390/biom13091343

**Published:** 2023-09-04

**Authors:** Krzysztof Jan Abram, Douglas McCloskey

**Affiliations:** 1Novo Nordisk Foundation Center for Biosustainability, Technical University of Denmark, 2800 Lyngby, Denmark; kabram@its.jnj.com; 2Johnson & Johnson MedTech, Bregnerodvej 133, 3460 Birkerod, Denmark; 3BioMed X Institute, Im Neuenheimer Feld 515, 69120 Heidelberg, Germany

**Keywords:** tandem mass spectrometry, deep learning, generative models, variational autoencoder, disentangled representation, latent space

## Abstract

Generative modeling and representation learning of tandem mass spectrometry data aim to learn an interpretable and instrument-agnostic digital representation of metabolites directly from MS/MS spectra. Interpretable and instrument-agnostic digital representations would facilitate comparisons of MS/MS spectra between instrument vendors and enable better and more accurate queries of large MS/MS spectra databases for metabolite identification. In this study, we apply generative modeling and representation learning using variational autoencoders to understand the extent to which tandem mass spectra can be disentangled into their factors of generation (e.g., collision energy, ionization mode, instrument type, etc.) with minimal prior knowledge of the factors. We find that variational autoencoders can disentangle tandem mass spectra data with the proper choice of hyperparameters into meaningful latent representations aligned with known factors of variation. We develop a two-step approach to facilitate the selection of models that are disentangled, which could be applied to other complex and high-dimensional data sets.

## 1. Introduction

Unsupervised representation learning of metabolites based on tandem mass spectrometry (MS/MS) data has important implications for the identification and structural elucidation of metabolites. The aim of unsupervised representation learning is to learn an interpretable and instrument-agnostic digital representation of metabolites directly from MS/MS spectra. Interpretable and instrument-agnostic digital representations would facilitate comparisons of MS/MS spectra between instrument vendors and enable better and more accurate queries of large MS/MS spectra databases for metabolite identification. Mass spectrometry is a predominant and comprehensive technique for metabolite identification owing to its high throughput and low sample requirements that rely on exact mass for compound identification. Mass spectrometry generates ionized fragments of a compound, separates those fragments by their mass-to-charge ratio (*m*/*z*), and measures the relative abundance of each fragment, resulting in a mass spectrum that can be used for identification and structural elucidation of metabolites. Many MS/MS parameters (referred to as factors) influence the ionization, fragmentation, and measurement process. These factors include ionization mode, collision energy, instrument type, and many more (see Figure 1). In particular, differences in the sample introduction method, ion source, fragmentation method, mass analyzer, and detector between instruments have a substantial influence on the final spectrum, which makes comparisons between spectrum acquired on different instruments problematic.

The majority of previous works have approached the problem of metabolite identification from MS/MS spectra by either comparing unknown spectra against large chemical libraries of known spectra, e.g., Global Natural Products Social Molecular Networking (GNPS) [1], The Human Metabolome Database (HMDB) [2], or MassBank of North America (MoNA) [3], or by attempting to learn a mapping from MS/MS spectra to metabolite, e.g., CSI:FingerID [4], MassGenie [5], or QC-GN2oMS2 [6], using machine learning. The former approach is limited by available metabolite standards and hampered by the differences in mass spectra obtained on different MS instruments. The latter approach is limited by the availability of labeled MS/MS spectra, which prevents generalization to underrepresented compound classes, such as natural products that are important for biomarker and drug discovery applications. Furthermore, algorithms derived from the former and latter approaches often fail to identify the correct compound [7].

In contrast, fewer works have approached the problem from the angle of generative modeling, where the aims are twofold: (1) to learn instrument-agnostic digital representations of metabolites directly from MS/MS spectra and (2) to learn generalizable rules for synthesizing MS/MS spectra from their digital representations. The most recent advances in deep generative modeling using variational autoencoders (VAE) [8] enable finding meaningful representations of complex data distributions in an unsupervised manner. It is assumed that a dataset X with independent and identically distributed (i.i.d.) samples x∈X is generated by an unknown random process, which includes latent variables *Z* that represent a set of independent generative factors. The VAE is trained to approximate the underlying high-dimensional and complex data distribution P(X) on the dataset X. A neural network fully parameterizes the target probability distribution; thus, finding the *P*(*X*) is posed as an optimization problem that can be solved using Stochastic Gradient Descent (SGD) [9]. After training on hundreds of thousands of samples x, the learned data distribution should enable the generation of novel samples, the interpolation between samples, and the classification of unknown samples. For mass spectra, this would enable the generation of new mass spectra of known metabolites or mass spectra of hypothetical metabolites, interpolating between the mass spectra of different metabolites or the same metabolites acquired on different instruments and with different instrument parameters and the classification of unknown metabolites based on their mass spectra.

A major challenge of generative modeling is to assign an interpretation to the latent variables *Z* in order to tell to which generative factors they correspond. For the case of MS/MS spectra, generative factors would include collision energy, ionization mode, instrument type, etc. Analogous to Principal Component Analysis, the goal is to find an independent or uncorrelated set of orthogonal components, where each component can be uniquely identified as the variation factor. In other words, we would like to model the latent space composed of latent variables Z in such a way that each orthogonal dimension of this space correlates with a single generative aspect of the data, often referred to in the literature as *disentanglement* [10,11]. Recent works have demonstrated the ability of VAEs, including BetaVAE [12] and JointVAE [13]. MNIST, dSprites, and Cars3D [14]. However, no previous work has explored the disentanglement of MS/MS spectra.

Previous work from our group utilized generative modeling to overcome the limitation of unlabeled MS/MS spectra by developing a multi-modal weakly supervised algorithm to learn a joint representation of MS/MS spectra and metabolite structures to enable learning from unpaired MS/MS spectra and metabolite structures [15,16]. While we showed that the joint representation could enable cross-modality reconstruction with only a single input modality, the joint representation was entangled, which prevented the utilization of MS/MS spectra from different instruments and the controlled synthesis of MS/MS spectra with specific generative factors.

In this study, we apply deep generative modeling with disentanglement to MS/MS spectra towards developing instrument-agnostic digital representations of metabolites from MS/MS spectra and controlled synthesis of MS/MS spectra. We first developed several proxy classification tasks to understand the dependencies and correlations between candidate MS/MS factors, including ionization mode, collision energy, instrument type, and others, to build an intuition for what MS/MS factors are independent and causal. We then trained BetaVAE and JointVAE models on the HMDB [2] and MoNA [3] data sets to obtain meaningful latent representations of MS/MS spectra. We qualitatively explored the learned latent representations through the conditional generation of novel MS/MS spectra, latent traversals and interpolations, and visualization. We quantitatively evaluated the latent space disentanglement using selected causal factors of MS/MS spectra variation and disentanglement metrics from the literature. Shortcomings of the published disentanglement metrics revealed when applying them to MS/MS spectra led us to develop a two-step pairwise correlation analysis to better automate the search for candidate generative models that exhibit disentanglement. Finally, we show that many of the causal MS/MS factors, such as ionization mode, collision energy, and instrument type, can be inferred directly from spectra using generative modeling. We provide the specvae package to preprocess MS/MS datasets, train and evaluate VAE models, perform operations in the latent space, and evaluate models with disentanglement metrics.

## 2. Methods

### 2.1. Experimental Design

This section describes the layout and purpose of each experiment performed in this study, along with the computational design. An overview of models and methods is included to provide context for understanding the experimental design. Further details of models and methods are expanded upon in their own sections further below.

The most common approach to representing MS/MS spectrum for machine learning tasks is binning of mass spectra [4,5,16,17]. Binning is most suitable for low-mass resolution instruments where discretization of mass up to 0.1 Da for large mass ranges can be used without incurring large memory overheads. However, binning is not suitable for high-mass resolution instruments where critical mass information is thrown away during the discretization process (e.g., isotope natural abundance ratios that can provide unambiguous chemical formulae) or spectrum mass ranges are limited (e.g., to less than 1000 Da precluding many larger natural products) due to memory limitations. We developed a sparse representation of the MS/MS spectrum consisting of vectorized pairs of mass and intensity values that overcome the limitations of binning by making use of the full instrument mass resolution and scan ranges while being highly memory efficient.

In Section 3.1, we validated the suitability of sparse representation of spectra by performing machine learning experiments that involved training many classification and regression models on MoNA and HMDB datasets to predict MS/MS parameters (also referred to as factors), using MS/MS spectrum and the remaining factors as input features.

We assessed the influence of the data preprocessing parameters, e.g., the maximum number of peaks allowed in the vector representation of a spectrum, the minimum intensity of an individual peak, etc. (see Section 2.5), as well as the BetaVAE model hyperparameters, e.g., regularization parameter beta, number of fully connected layers in the network, etc., on the reconstruction quality. The objective function of the BetaVAE model can be divided into two elements. The first is the reconstruction term, which encourages the model to learn latent representations z from the data samples x and reconstruct them accurately as x’, and the second is the KL divergence term, which encourages the inferred posterior distribution to match the prior distribution. Beta is a regularization parameter that changes the proportion of divergence term in the loss function, and if that value is greater than 1, it prioritizes learning disentangled representations of the input samples. In this experiment, we explored only the reconstruction term of the BetaVAE objective function; specifically, we investigated how model and data hyperparameters affected the reconstruction quality of the BetaVAE model. We used 4 metrics to evaluate the reconstruction quality of the data samples: cosine similarity, Euclidean distance, percentage change, and percentage difference. We used average reconstruction scores for the baseline metrics (see Section 2.3).

In Section 3.2, we trained classification and regression networks jointly with the Variational Autoencoder. The input for the BetaVAE model was the MS/MS spectrum, which was first compressed by the contracting encoder network to create a low-dimensional representation of the spectrum, also known as the latent representation, which was subsequently decompressed back to the spectrum by the decoder network which reconstructs the input spectrum. The compressed encoding of a spectrum was simultaneously passed to the downstream classification/regression model, and similarly to the previous experiments that involved predicting a factor associated with the MS/MS spectrum from the dataset, the input consisted of the compressed encoding of a spectrum (instead of the spectrum) and other factors from the dataset. The aim of this experiment was to verify the assertion that the latent representation of the MS/MS spectrum carries the same information as the spectrum; hence, the prediction performance achieved in the latent space should be comparable to those performed in the input space. Similar to the previous experiment, we analyzed the performance and PFI of the downstream models, and we compared the outcome with the corresponding models trained only on spectra.

In Section 3.3, we investigated the principal relations between 10 factors associated with each MS/MS spectrum in the MoNA dataset, e.g., collision energy, instrument type, etc. On that account, we cross-correlated these factors to observe whether any pair was correlated and, if so, to what degree.

Next, we trained a set of classification and regression networks that predicted a factor from the dataset. The input to each model was a vector composed of an MS/MS spectrum concatenated with all remaining factors from the dataset, except for the factor that we used to predict, referred to as the target factor. Consequently, we had 7 classification and 3 regression task categories, depending on whether the factor was modeled as a continuous or discrete variable, e.g., a classifier that predicted the ionization mode from the spectrum and 9 remaining features, a regressor that predicted collision energy value from the spectrum and other 9 remaining features, and so on. In addition, we tuned the data preprocessing parameters for each task category to observe the impact of these parameters on the performance of the models.

Further, we trained a set of classification and regression networks jointly with the BetaVAE on the latent encoding of the spectra instead of the spectra themselves. Similar to the previous settings, the input to each model was a vector composed of a latent encoding of the MS/MS spectrum concatenated with all remaining factors from the dataset.

And finally, we analyzed trained models with regard to their performance and the permutation feature importance (PFI) [18]. The latter metric provided us with a notion of the relative importance of the feature with respect to the target variable in the classification or regression task, i.e., to what extent the given input feature impacted the accuracy of the model, e.g., while predicting the ionization mode, what was the impact of the collision energy or precursor type on that prediction.

The above strategies provided us an opportunity to compare model performances when trained on spectra against those trained in the latent space of the BetaVAE model. In the context of the feature importance, it provided us with the ability to observe whether the latent space of the BetaVAE promoted certain factors in the classification/regression more than others and see what these factors were.

In Section 3.4, we investigated the impact of the regularization factor beta on the points in the latent space of the BetaVAE model. The goal of the analyses was to provide insights into the latent space of the selected BetaVAE models trained on the MS/MS data, i.e., how data points were located in the latent space and how they related to the MS/MS factors, e.g., total exact mass, collision energy, etc. Visualization methods were used to build intuition about the structure of the latent space for a subset of the MoNA dataset by encoding spectra to the latent space using the encoder network and then projecting the latent space into a reduced 2D space using PCA. Moreover, we investigated if points in the latent space were distributed according to continuous factors, specifically total exact mass and collision energy. We examined if they could be clustered by any of the categorical factors, especially instrument type.

In Section 3.5, we investigated the shape of BetaVAE and JointVAE latent spaces. We used a two-step latent traversal method to qualitatively evaluate the newly generated samples and investigate the latent space of the BetaVAE and JointVAE. First, the latent points were sampled evenly across the selected line or plane from the set of latent dimensions; second, the points were fed to the decoder network to generate the spectra). We chose to sample points in the 2D plane after fixing the third dimension in order to create a 2D grid visualization of generated spectra, which provided better intuition about the quality and shape of the latent space.

In Section 3.6, we investigated the semantics of the latent space through interpolation. In the previous experiment, we showed examples of traversal across orthogonal directions parallel to the main axes of the latent space. However, in the high-dimensional continuous latent spaces, the linear interpolation might sample points from regions that are extremely unlikely given the Gaussian prior. Alternatively, the spherical linear interpolation (slerp), i.e., a path on the great circle of an N-dimensional sphere that corresponds to an arc linking two points, is more likely to sample in the proximity of high probability [19]. Slerp was used to verify the BetaVAE model’s generalization ability to synthesize valid spectra not found in the training dataset. The method developed allows one to take advantage of the continuity of the chemical space, in this case, using only the mass spectrum modality of the chemical compounds and sample novel spectra.

In Section 3.7, we investigated the suitability of current disentanglement metrics on model selection. The best metrics to measure disentanglement in generative models is an open and active area of research that does not yet have a definitive conclusion. Therefore, due to differences in definitions of the disentangled representations and the contention between researchers on how to quantify the disentanglement, we used 3 different disentanglement metrics that have been widely cited by the community to quantify the degree to which selected models achieved disentangled representations: BetaVAE [12], FactorVAE [20], and MIG [21]. In our study, we evaluated metrics on MoNA and HMDB datasets and BetaVAE and JointVAE model types. First, we analyzed how much the existing disentanglement scores agreed and how much they varied across different model/preprocessing parameters. This analysis allowed us to address how important different hyperparameters were for disentanglement. Second, we analyzed whether disentanglement metrics could be used for model selection. Specifically, we focus on whether we could reliably and consistently distinguish bad and good models using the disentanglement metrics. Note that while running our experiments, we separated the impact of the regularization factor (beta in the BetaVAE model, gamma in the JointVAE models), the maximum number of peaks, and model architecture from other inductive biases, e.g., learning rate, epochs, optimizer, normalization, etc. This approach allowed us to limit the number of variables that would impact those scores but had no intrinsic value with reference to the actual representation.

In Section 3.8, we developed an approach to uncover the factors of variations for MS/MS spectra encoded in the latent space. Unlike MNIST or other machine vision datasets, it is difficult to identify the true underlying factors of variation in mass spectrometry as the data are too complicated and high-dimensional to easily visualize and compare between samples. Therefore, we sought to develop an approach for better determining if select factors have been disentangled without the need for visual intuition. Given that the selected attributes are true factors of variation and that the BetaVAE model, in fact, achieves the disentanglement, we expect these factors to correlate with the latent variables, i.e., a coordinate Z_i_ of the low-dimensional latent representation Z of the spectrum. We found this method to have special applications for modalities such as mass spectra where the factors cannot be observed directly in the input space. In the correlation analysis, devised to test whether the disentanglement is true or not with reference to the assumed factors, each factor F_i_ must correlate with a distinct latent variable Z_j_. Note that *distinct* means that each factor correlates only with one latent variable exclusively (i.e., no other factors correlate with the same latent variable). In an ideal example, each factor F_i_ would correlate strongly with only one distinct latent variable Z_j_, and the same factor would have a very weak or absent correlation with the remaining latent variables.

Next, we determined how consistent the above observation was across a subset of the MoNA dataset, and we summarized results in the form of distributions of correlation coefficients. The method developed provided us with tools to compute a comprehensive overview of all cases, helped us to conclude which factor correlated with the latent variable stronger, and provided insights into our notion of distinctiveness.

### 2.2. Model Design and Architecture

The classification and regression tasks used deep neural networks. Depending on the target variable and the input features, the input and the output of the models varied in size. The number of intermediate layers also varied from model to model. Therefore, the exact number and sizes of layers in the model were included in the Appendix A along with other model/training parameters and resulting metrics. Each layer was a fully connected layer followed by a batch normalization [22] and nonlinear activation function ReLU, with the only exception for the output layer, which did not use batch normalization and consisted of a linear activation. All classification and regression models obeyed the same principle with regard to how the model was constructed.

Other experiments used the Variational Autoencoder (VAE) [8], specifically two modifications of the basic framework, i.e., BetaVAE and JointVAE [12,13]. The VAE model is a deep generative model (DGM), which is able to approximate the underlying high-dimensional and complex data distribution P(X) on the given dataset X with the i.i.d. samples x. The neural network fully parameterizes the target probability distribution; thus, finding the P(X) is posed as the optimization problem that can be solved with stochastic gradient descent (SGD) [9,23]. The VAE model consists of the encoder network that approximates the posterior probability q(z|x) and the decoder network that approximates the likelihood probability p(x|z). The VAE optimizes the evidence lower bound, also known as the ELBO [9].

The BetaVAE model introduces a hyperparameter beta that modifies the proportion of the Kullback–Leibler divergence component in the VAE loss function. In the special case when beta = 1, the BetaVAE reduces to the VAE model. The parameter beta is a regularization factor: when beta > 1, the network emphasizes the KL divergence more, exerting pressure on the posterior q(z|x) to match the prior p(z). This, however, comes at the cost of reconstruction quality, i.e., the bigger the beta value, the worse the reconstruction quality (Figure 4) [12]. The BetaVAE automatically discovers independent and interpretable continuous latent representations in a completely unsupervised manner [24], and the JointVAE extends this idea to model both continuous and discrete latent variables (at least for synthetic computer vision benchmark datasets).

The JointVAE framework learns disentangled continuous and discrete representations, i.e., it allows one to model both continuous and categorical factors. The loss function incorporates discrete and continuous KL divergence terms. However, direct optimization of the objective function often ignores the discrete component. The solution to this particular problem is to introduce a capacity parameter that is separate for the discrete and continuous channels, which controls the amount of information carried in each channel. As the training progresses, the capacity grows and encourages the gradual exploration of factors [13].

Our implementation of the VAE models was similar to the classification and regression models, parameterized model architecture by the number of layers and their sizes as a model configuration. The VAE models consisted of a top–down encoder network and a bottom–up inference decoder network. Both encoder and decoder layers were fully connected, followed by batch normalization and nonlinear activation function ReLU. The exceptions were the last layer of the encoder, where no activation function was applied, and the output layer of the decoder, where the activation function was sigmoid and no batch normalization was used.

In tasks that involved training a downstream classification/regression model using the latent encodings of the spectra and simultaneously training the BetaVAE model, referred to before as joint training, we combined the loss functions of the downstream predictive model and the upstream autoencoder. Specifically, the loss function was a sum of the BetaVAE objective and the downstream model objective, e.g., in the case of the regressor as the downstream model, the new objective was a sum of the MSE value and the BetaVAE loss value. The architecture of the downstream classification/regression and the BetaVAE models followed the same logic as described above.

### 2.3. Qualitative and Quantitative Approach to Validate Models

In order to systematically quantify the performance of numerous training tasks and to be able to effectively compare resulting models, we used appropriate metrics as described in the next sections. We divided our evaluation approach into three major categories, i.e., supervised learning, spectra reconstruction, and disentanglement analysis.

#### 2.3.1. Assessment of Supervised Models

In all supervised learning tasks, such as classification and regression, we used standard metrics to evaluate models. For classification models, accuracy, balanced accuracy, recall macro, precision macro, and F1 score were used. For regression models, MSE, RMSE, MAE, R^2^ (coefficient of determination), and explained variance score were used. Additionally, we used permutation feature importance (PFI) to assess the influence of the features on the model prediction. This score revealed how much information was carried within a particular feature, consequently revealing how much it contributed to the prediction.

#### 2.3.2. Reconstruction

Evaluating VAE models can be broken down into two categories: assessing reconstruction quality and evaluating disentanglement. For the reconstruction, we use 4 different metrics, i.e., cosine similarity (cos_sim—ngular difference), Euclidean distance (eu_dist—magnitude difference), percentage change, and percentage (per_dif—absolute and relative change in value). The initial analysis of the reconstruction scores gave rise to concerns about the ambiguity of the values of the 4 different metrics. Specifically, none of the metrics revealed how a particular reconstruction output deviated from the average reconstruction, i.e., in the case when the autoencoder reconstructs averages of the samples in the dataset. 

To establish a baseline value for each metric in the reconstruction task, we propose a new metric, namely average reconstruction score, that depends solely upon the data preprocessing parameters. Consequently, the average reconstruction score is not a model evaluation metric but a baseline metric. The definition of the metric is the following: given a dataset, X with samples x∈X, find the average sample $\underline{x}$ and compute
(1)$$R_{avg}^{score}=\frac{1}{n}\sum_{i=1}^{N}score(x_{i},\underline{x}),$$
where the *score* is any of the 4 above-listed metrics. Finally, the input spectrum and reconstructed spectrum were plotted side by side for a visual comparison for qualitative reconstruction assessment.

#### 2.3.3. Disentanglement

The most recent advances in learning disentangled representations allowed us to verify the degree to which models were able to find such representations or states with a degree of certainty otherwise. Visualization can be used as a first approximation to qualitatively assess the degree of disentanglement. Given that the inference network was trained to reconstruct samples well, i.e., the model had a small reconstruction error, we can select a set of latent points, reconstruct them with the decoder network, and compare results by plotting obtained novel data samples. Moreover, to build an intuition about the latent space, we traversed linearly through the latent space by mutating a single coordinate and keeping the remaining coordinates fixed. If the reconstructed samples showed a gradual change of a single factor, then the latent representation was considered disentangled.

While visualization is an adequate technique in the analysis only if the data samples are images, it might be inappropriate for all modalities. Another disadvantage is that the visualization does not provide any tools for automated search for models that might exhibit the desired property. Disentanglement metrics (DM) are designed to overcome this disadvantage by quantifying the disentanglement as the numerical score. In this work, we considered three scores: (1) the BetaVAE metric [12] captures disentanglement as the accuracy of a classifier that predicts the index of a generative factor; (2) FactorVAE [20], an improvement to the previous metric, uses majority vote classifier and accounts for the edge case in the BetaVAE metric; and (3) Mutual Information Gap (MIG) [21], the information-theoretic score, for each factor of variation, measures the normalized gap in mutual information between the highest and the second-highest coordinate in latent representation.

### 2.4. Datasets and Feature Selection

In this study, we considered two tandem mass spectra (MS/MS) datasets: MoNA and HMDB [2,3]. The first dataset is composed of fragmentation spectra from the MoNA library and incorporates multiple factors of variation. Specifically, a combination of instrument types, range of collision energies, compound mass, and positive or negative ionization modes. The dataset was composed of 125,831 samples, whereby 86,381 were in positive ionization mode, and 38,965 were in negative ionization mode. Altogether, the dataset represented 15,956 compounds with a unique InChIKey identifier. The MoNA dataset is also extremely unbalanced with respect to instrument type category, as depicted in Figure 1C. The major 6 distinct instrument classes constitute around 92% of the entire dataset, while the remaining 34 classes make up merely 8%.

The second dataset was picked from the HMDB library and represented fewer factors of variation. Specifically, compared to its previous counterpart, all 92,916 fragmentation spectra corresponded to a single instrument type. The dataset did not include other features, e.g., total exact mass, precursor type, precursor *m*/*z*, etc. Nevertheless, it is perfectly balanced for the ionization mode category, and collision energy values selected collision energies and were chosen. The dataset represents 15,486 unique metabolites with 6 different spectra for each compound. There are 3 spectra per ionization mode, within which each spectrum has a distinct collision energy value, i.e., 10, 20, and 40.

Both of the sets are additionally supplemented with the compound classification metadata, i.e., kingdom, superclass, class, and subclass, provided by the ClassyFire computational tool [25].

### 2.5. Data Processing and Spectrum Representation

The MS/MS spectrum consists of arbitrarily many peaks located on the continuous mass-to-charge (*m*/*z*) axis. Therefore, an accurate representation of the spectrum has an inherently sparse nature. For example, spectra in the MoNA dataset had, on average, 260 peaks, with a median of 33 peaks, a minimum of 1 peak, and a maximum of 110,235 peaks. In the case of the HMDB dataset, spectra had roughly 28 peaks on average, with a median of 31 peaks, a minimum of 1 peak, and a maximum of 31 peaks.

Many machine learning models can benefit from having their input converted into a dense, vectorized form. Several preprocessing operations for modeling mass fragmentation spectra were taken and can be broken down into two categories: (1) filters that alter the inner structure of the spectrum, i.e., accept or reject peaks according to specified conditions, and (2) transformations that modify the internal values of the spectrum, without changing its structure. One filter was the minimum and the maximum possible value for the *m*/*z*, limiting the mass range. Another filter imposed a minimum peak intensity threshold. We assumed that peaks of a larger relative abundance are more likely to carry more information than peaks with a smaller relative abundance and will more likely be conserved across instruments, instrument settings, and sample matrices. We sorted peaks by their intensity value in descending order and took the top N peaks. The latter filter brings all MS/MS spectra to the same maximum length, i.e., the number of peaks per sample.

We used two transformations, one that projected intensities and *m*/*z* values into the range [0, 1] and another that used dynamic range expansion. Specifically, our preprocessing workflow was the following: First, we defined the following filtering operations:Reject peaks below the threshold min_intensity parameter;Reject peaks with *m*/*z* values above the max_mz parameter;Limit the number of peaks in the spectrum to N = max_num_peaks top intensity peaks, i.e., sort peaks by the intensity value in the descending order and take the first N instances.

Second, we applied the following transformations:Project intensity and *m*/*z* values into the range [0, 1];Normalize intensity of peaks, i.e., rescale all peaks into the range [min_intensity, 100]. If there is only one peak, no rescaling is applied.

Finally, a spectrum was represented as a vector. Each peak was structured as a 2D vector where the first dimension was the *m*/*z* and the second dimension was the intensity value. To form a full spectrum, the peak vectors were concatenated in descending order with reference to intensity. The spectrum with N peaks was encoded as a vector with 2N cells. Figure 1A depicts the spectra preprocessing pipeline. This representation of spectra allows for expressing *m*/*z* and intensity accurately with the real numbers. Recently, a similar dense spectral representation was applied to the analysis of proteomics data [26].

### 2.6. Model Training and Hyperparameter Tuning

All experiments involving training deep neural networks utilized the Adam optimizer with a learning rate of 1 × 10^−3^. While training and evaluating classification models, we considered class imbalances in our datasets. To minimize the impact of class disproportions, we used a weighted random sampler approach, which ensured that a batch consisted of a similar number of instances from each class, minimizing the class disparity at the batch level. This method provided better results consistently for binary and multi-class classification tasks, even if the ratio of minor class to major class was smaller than 0.1. Other methods, such as class weights, which is a method that modifies the loss function behavior by weighting the loss of smaller classes in the case of cross-entropy, often cause unstable training observed as jittering loss function across epochs.

We used GPU implementations of metrics while evaluating our models; this way, we minimized the data transfer between GPU and CPU devices, especially in cases when the training ran entirely on the GPU. Metrics such as permutation feature importance and correlation scores were computed on the CPU. All models were trained using Intel Xeon Gold Series CPU, 500 GB of RAM, and 4 GPUs NVidia Tesla A100 with a total of 160 GB of memory. Altogether, we trained over 16,180 models, worth 2 months of continuous training: 600 classifiers, 230 regressors, and 630 classifiers trained jointly with the VAE model; 380 regressors trained jointly with the VAE model; 1220 BetaVAE models; 10,680 BetaVAE models with lower capacity; and 2430 JointVAE models.

## 3. Results

### 3.1. Sparse MS/MS Spectra Representation Improved Memory Efficiency without Compromising Machine Learning Performance

In this study, we developed a sparse representation of the MS/MS spectrum consisting of vectorized pairs of mass and intensity values as the memory-efficient alternative to binning representation, which trades mass resolution and range for memory [4,5,16,17].

To validate the suitability of our sparse representation for machine learning, supervised machine learning tasks were constructed, which involved training many classification and regression models on MoNA and HMDB datasets to predict MS/MS parameters (also referred to as factors) using MS/MS spectrum and the remaining factors as input features. Specifically, the input of the predictive model was composed of a spectrum vector using our sparse representation and a factor vector concatenated into a single input vector. For example, a classification model that predicts ionization mode has input consisting of a spectrum vector and the following factors: collision energy, total exact mass, precursor *m*/*z*, precursor type, instrument, instrument type, kingdom, superclass, and class.

We tested the Cartesian product of the preprocessing parameters: max_num_peaks with values 10, 15, 25, 50, and 100; min_intensity with values 0.001, 0.01, 0.1, 0.2, 0.5, and 1.0; and rescale_intensity set to True and False (with and without dynamic range expansion). Altogether, we trained 604 classification models and 227 regression models. The results can be found in Appendix A for classification models and Appendix A for regression models. In general, we found that the evaluation metrics for regression and classification models did not change significantly depending on the preprocessing parameters used. Among all the preprocessing parameters, the maximum number of peaks, i.e., N = max_num_peaks, had the largest effect on the final scores in both classification and regression (only when predicting the total exact mass) tasks. However, increasing the maximum number of peaks showed minor improvements in classification model accuracy and a moderate decrease in the regression model error.

We further tested the suitability of the sparse spectrum representation in the context of generative modeling. The data preprocessing parameters analyzed included the number of peaks used in the spectra representation (max_num_peaks), the cut-off value for peak intensity values (min_intensity), and the dynamic range expansion for peak intensity values (rescale_intensity) (Figure 2). We observed that higher values of the number of peaks consistently showed lower performance in the reconstruction task, which indicated that the size of the spectra representation was negatively correlated with the reconstruction quality on average (Figure 2, third row). Related, we observed that lower values of the min_intensity also consistently showed lower performance in the reconstruction task by limiting the size of the spectra representation (Figure 2, fourth row). The dynamic range expansion, i.e., rescale_intensity, did not affect results. These results indicate that the sparse spectrum representation can be modulated to capture greater levels of complexity and details as required for downstream tasks.

We also evaluated the impact of multiple model hyperparameters on the reconstruction scores (Figure 2). The model hyperparameters included model architecture, symmetry of the encoder and decoder networks, size of the latent representation, and influence of the regularization strength beta. We found that certain hyperparameters had a larger impact on scores than other parameters when evaluated on the MoNA dataset (Figure 2). Specifically, the size of the latent dimension was correlated positively with the cosine similarity and correlated negatively with the Euclidean distance and percentage difference (Figure 2, first row). This suggests that larger latent representations are easier to reconstruct by the decoder network. Also, very shallow models, i.e., model_depth = 0, had weak performance compared to models with model_depth > 0 (Figure 2, second row) but did not show any improvements compared to the models with model_depth = 1. The symmetricity or asymmetricity of the encoder and decoder network had no impact on the final scores, regardless of the remaining hyperparameters. In general, beta was negatively correlated with cosine similarity and positively correlated with Euclidean distance and percentage difference, meaning that a higher value of the regularization factor resulted in lower reconstruction quality.

### 3.2. The MS/MS Factor Inference Based on Latent Encodings Was Superior to the Inference Based on Spectra

Dimensionality reduction is frequently used in machine learning as a feature extraction method. In most cases, complex data can be represented using a smaller number of parameters compared to the original representation, which simplifies the analysis of the data. A reduced representation also enhances the performance of machine learning algorithms by reducing the complexity of models, which leads to better generalization and prevents overfitting on training data. The quality of the compressed representation of spectra was evaluated by comparing the performance of models that predict associated factors, e.g., ionization mode, directly from raw spectra, and the performance of models that predict factors from the latent representations. Here, we used the BetaVAE model to generate the latent representations (see Section 2). Appendix A contain the detailed results of the training models used in these experiments.

When applied to the MoNA and HMDB datasets, we found that the inference of each factor (on the *x*-axis) performed on the latent representation of spectra was superior to the inference on spectra alone for both datasets (Figure 3, Appendix A). We observed that increasing the regularization parameter of the BetaVAE model negatively impacted classification and regression performance, and the best results were always obtained for the minimal value of beta = 0.01. On the other hand, increasing the BetaVAE latent dimension positively impacted classification and regression performance, and the best results were consistently better for the largest latent dimension tested. The relation between regularization strength and latent dimensions size indicates a trade-off between retaining information in the latent space and matching the independent Gaussian mixture prior to training.

### 3.3. Feature Importance Analysis Revealed Relative Importance and Dependencies between MS/MS Factors

Current methods for representation learning using generative modeling require a hypothesis of the factors of generation. While domain knowledge contributes to the hypothesis generation, we sought to recover the factors of MS/MS spectrum generation in a less biased fashion by constructing supervised classification and regression tasks that would reveal the most causal factors and their correlations. One way to identify causal factors in machine learning is through feature importance analysis. The feature importance analysis is a method to quantify the effect of the input features on the model performance by scoring each feature according to its contribution to the model prediction. We used permutation feature importance (PFI) in this work, a model-agnostic feature importance score. The permutation feature importance measures the decrease of the model performance while permuting an input feature. The feature is unimportant if shuffling does not change the score, and conversely, the feature is important when permuting that feature causes a decrease in the model performance [18,27].

We computed the PFI score for each classification and regression model described previously to understand which MS/MS factors had the largest impact on the model performance. For example, in the classification task to predict ionization mode, the top 4 important features were instrument type, collision energy, precursor *m*/*z*, and total exact mass (only MoNA) (Figure 4E). In the regression task to predict collision energy, the top 4 important features were the spectrum, instrument, instrument type, and total exact mass for the MoNA data set and spectrum and ionization mode for the HMDB data set (Figure 4E). We observed a shift in feature importance when including either the raw MS/MS spectrum or the learned latent representation as input (Figure 4A,B). We also observed the relative PFI changes in instrument, precursor type, superclass, and precursor *m*/*z* presented in Figure 4A,B to be robust to changes in hyperparameters and architectures. The shift in feature importance from raw MS/MS spectrum to learned latent representation indicates that certain MS/MS factors are better captured in the latent space than others. Detailed results for each model can be found in Appendix A.

In addition, we also explored how BetaVAE model hyperparameters influenced the PFI scores to quantitatively determine how modeling choices modulated the information retained in the latent space. We observed that the latent_dim parameter had a major impact on the PFI scores. For example, the lower latent dimensions (latent_dim = 3 and 4) emphasized the instrument factor over the latent spectrum, and conversely, the higher latent dimensions (latent_dim = 10 and 20) emphasized the latent spectrum over the instrument factor when predicting collision energy (MoNA). In another example, the lower latent dimensions (latent_dim = 3 and 4) pushed the class factor importance high above the superclass score, while the higher latent dimensions (latent_dim = 10 and 20) reversed that order for the MoNA dataset, while the higher latent dimensions pushed spectrum (latent) importance scores up while decreasing importance scores for class and superclass when predicting ionization mode. For the remaining MS/MS factors as the target factors, the impact of the latent_dim parameter was smaller or unobservable. The beta parameter, on the other hand, minimally affected the PFI scores. For all target factors, the beta parameter showed no significant influence on final PFI scores in the MoNA dataset, while the higher *beta* values entailed lower importance scores and did not cause any changes in the ranking of input factors in the HMDB dataset.

While feature importance can indicate candidate causal factors, it does not reveal correlations between factors. We cross-correlated MS/MS factors for MoNA and HMDB datasets using Pearson’s Correlation Coefficient method to identify correlated factors that would be redundant in downstream representation learning (Figure 4C). For example, the total exact mass and precursor *m*/*z* were strongly correlated with a score of 0.96, instrument and instrument type were moderately correlated with a score of 0.39, and all other pairs had close to zero correlation in the MoNA dataset.

### 3.4. Qualitative Analysis of the BetaVAE Latent Space

After building an intuition for the causal factors in MS/MS spectrum generation described in the previous sections, we qualitatively explored the structure and contribution of select factors to the composition of the BetaVAE latent space using principal component dimensionality reduction, clustering, and visualization. Visualization revealed that as we increased the beta parameter, the variance proportion of PC2 decreased until beta = 2.0, where the points were compressed to a linear form, and the entire variance was dominated by PC1. Similarly, we noted a change in the mass distribution where the low and high-mass compounds were clearly distinguishable for beta values 0.01, 0.1, and 0.2 but were hard to tell apart for larger values (Figure 5A).

Next, we observed that when points in the latent space were colored according to total exact mass or collision energy (Figure 6A,B), the density of points could be demarcated into regions of greater or lower values starting at the origin and radiating outward with smooth transitions between them (Figure 6C,D). Furthermore, the regions of high total exact mass and high collision energy were inversely correlated, whereby the origin was populated with points that contained low mass and high collision energy spectra, and the periphery was populated with points that contained high mass and low collision energy spectra (Figure 6C,D).

Last, we qualitatively observed clustering by instrument type, instrument vendor, and superclass (compound taxonomy, Figure 5B(i–iii)). Standard cluster performance metrics were used to quantify the accuracy of clustering (Figure 5B(i–iii) bar charts insets). We observed stronger clustering accuracy for instrument type and vendor compared to superclass, indicating that this particular model was better able to disentangle instrument attributes as opposed to chemical attributes of the spectra. Overall, the qualitative analysis of the latent space led us to conclude that the generative models BetaVAE and JointVAE with relatively simple architectures were capable of capturing factors of spectra variation in their latent space representations.

### 3.5. Traversal of BetaVAE and JointVAE Latent Space

We used a two-step latent traversal method to qualitatively evaluate the newly generated samples and investigate the latent space of the BetaVAE (Figure 7) and JointVAE (Appendix A). First, the latent points were sampled evenly across the selected line or plane from the set of latent dimensions (Figure 7A); second, the points were fed to the decoder network to generate the spectra (Figure 7B). We chose to sample points in the 2D plane after fixing the third dimension in order to create a 2D grid visualization of generated spectra, which provided better intuition about the quality and shape of the latent space (Figure 7C). The 2D grid visualization obtained through latent space traversals allowed us to verify that the model captured the samples’ variability in the training dataset (Figure 7D(i–iii)). Expectedly, we observed peaks at different *m*/*z* positions while the intensity of corresponding peaks changed as we traversed the latent space. Moreover, since the underlying latent points were evenly spaced, and the distance step was relatively small, the grid plots enabled us to observe that the adjacent spectra in the grid were similar while points further away were incrementally different, which would be expected if the latent space was smooth and continuous (Figure 7D). In contrast, while traversing along the disjoint discrete latent dimensions of the JointVAE, we did not observe a gradual transition in spectra plots; the adjacent spectra differed substantially as anticipated (Appendix A). However, while performing similar analyses for many models and a broad range of latent points, we found it difficult to align the underlying factors of variation with a given latent dimension. This was not unexpected as spectra are complex plots that are difficult to discern without the aid of informatics assistance, even for domain experts.

### 3.6. Interpolating between Spectra Using the Continuous Latent Space of BetaVAE Model

Biliverdin is a compound in the MoNA dataset that has multiple spectra representations acquired by different instrument types with varying collision energies and ionization modes (Figure 8B). When encoded by the BetaVAE encoder, the resulting latent points were distributed in the latent space in such a way that the collision energy of compounds and the first PCA component X1 were negatively correlated (Figure 8A). We selected points that corresponded to distinct instrument types (Figure 8A point no. 4+ LC-ESI-QTOF and 7-ESI-QFT), and we used the slerp method to sample points in the latent space (Figure 8A, the bold blue line). One would expect that interpolating between different instrument types would yield different spectra of the same or structurally similar compound. Subsequently, the set of points was decoded using the BetaVAE inference network, and the interpolants were compared against the entire database using the cosine similarity, Euclidean distance, and percentage difference as the similarity criteria (Figure 8C). The matches were sorted by the cosine similarity metric, and the best results were used for verification. Note that matches for Biliverdin should not be found as there were no Biliverdin spectra in the database acquired with parameters that would likely lie in the interpolated region. However, we would expect to see structurally similar compounds to Biliverdin, whereby spectra with a higher cosine similarity should also be more structurally similar. Indeed, we observe that the least structurally similar compound, Stenothricin H, has the lowest cosine similarity to Biliverdin, and the most structurally similar compounds, C36H56O9 and Colchicine, have the highest cosine similarity to Biliverdin.

When using the inference network of the BetaVAE model in the spectra comparison task, the reconstruction quality of the network plays a significant role since an inferior reconstruction quality leads to ambiguity in the spectra comparison, which may result in many likely matches. Therefore, to quantify the model’s ability to produce valid samples, we compared a reconstructed spectrum of a compound against the entire database using the similarity function, and then the position of the corresponding ground-truth spectrum on the sorted list with reference to the metric was used to rank the similarity value of that compound. In principle, the ranked score can be constructed for any similarity function, shown in Figure 8D, in which we compared three aforementioned metrics. We emphasize the fact that the choice of the similarity metric introduces an inductive bias. Therefore, we show the percentage difference, for which the median rank was 304, and the cosine similarity, for which the median rank was a lot better at 10 in the figure. The Euclidean distance and the cosine similarity were performed identically by giving the same rank ordering (Table 1).

### 3.7. Automatic Evaluation of Disentanglement Was an Unreliable Method for Model Selection When Applied to MS/MS Spectrum

The most recent advances in learning disentangled representations allowed us to quantify the degree to which models were able to find disentangled representations. Visualization can be used as a first approximation to qualitatively assess the degree of disentanglement but is not suitable for automated search for models that might exhibit the properties of disentanglement. Disentanglement metrics are designed to better automate model selection by quantifying the degree of disentanglement in a score [28]. We considered three scores: (1) the BetaVAE metric [12] captures disentanglement as the accuracy of a classifier that predicts the index of a generative factor; (2) FactorVAE [20], an improvement to the previous metric, uses a majority vote classifier and accounts for the edge case in the BetaVAE metric; and (3) Mutual Information Gap (MIG) [21], the information-theoretic score, for each factor of variation, measures the normalized gap in mutual information between the highest and the second-highest coordinate in latent representation.

When applied to MS/MS spectra, we found that the disentanglement metrics were not correlated, regardless of the beta value, indicating a weak agreement between them (Appendix A). The repeated plots for different beta values in Appendix A indicated the impact of the regularization on the relationship between metrics. For example, for small values of beta, scores were equally scattered regardless of the max_num_peaks parameter, but for larger values of beta, scores formed disjoint clusters, clearly separated by the grouping parameter. However, the grouping happened only with reference to the BetaVAE score, while other metrics did not exhibit the same behavior.

We evaluated around 700 BetaVAE models trained on the MoNA dataset with different hyperparameters, including max_num_peaks, beta, and model architecture, to investigate how hyperparameters affected disentanglement metrics. Figure 9 shows the distributions of scores for beta and max_num_peaks hyperparameters. FactorVAE and MIG had a minimal variance in the distribution of their scores (Figure 9A–C). In contrast, BetaVAE scores showed variation in response to different hyperparameter settings, and in particular, we observed that the parameter contributing the greatest variance to the score was max_num_peaks (Figure 9A,D). We observed that models trained on spectra representations that included more peaks obtained higher BetaVAE scores on average (Figure 9A). We can account for the correlation between the number of peaks and BetaVAE score due to the fact that the information that can be deduced about a particular compound is often increased when there are more peaks in the spectrum. This meant that models trained on a larger number of peaks were more likely to disentangle spectra better, and conversely, models trained on a smaller number of peaks were less likely to disentangle spectra. In addition, we observed that the scatter of these clusters decreased as the beta value increased (Figure 9A). We account for the inverse correlation between clusters and beta values through the inner workings of the BetaVAE score, which captures the disentanglement as the accuracy of a classifier that predicts the index of a generative factor. For high values of beta, data points represented in the latent space were less scattered and usually formed a thin plane, which caused the classifier to have a similar performance across different models, regardless of other hyperparameters.

To directly quantify the explanatory power of each hyperparameter towards a given disentanglement metric, we used a simple least square regression method to predict disentanglement scores using hyperparameters (Figure 9G). For the BetaVAE metric, if the regression results depended only on the *beta* parameter, we were able to explain 34% of the variance; similarly, the max_num_peaks parameter explained 43% of the variance, and only 0.5% came from the model architecture. The Cartesian product of all three parameters explained 78% of the variance. For the FactorVAE and MIG scores, only 17% and 12 % of the variance can be explained by hyperparameters, indicating that randomness dominated the values of these scores. The analysis was repeated for the JointVAE models (Figure 9H). While taking all five hyperparameters jointly in a similar regression task, we were able to explain ~11% of the BetaVAE score variance, ~4% of the FactorVAE score variance, and 23% of the MIG score variance.

### 3.8. Disentangled Representation for Fragmentation Mass Spectra Using Ground-Truth Factors of Variation

Based on our previous analysis of factor importance, we appointed three attributes from the MoNA dataset as the candidates for the factors of variations: (F_1_) instrument type, (F_2_) total exact mass, and (F_3_) collision energy. (F_1_) instrument type was chosen because different instrument types produced varied mass spectra for the same compound. (F_2_) total exact mass was chosen as the mass spectra contained the parameter value directly in its representation, which the model should be able to deduce. (F_3_) collision energy was chosen because it was considered the most impactful parameter on the resulting spectra: at low collision energy, the compound is fragmented into fewer and often larger masses, and at higher energies, the compound is fragmented into more but often lower masses. This directly translates into a variance in obtained spectra, i.e., low energy results in a few peaks at higher masses, and higher energy leads to spectra with more peaks at lower masses. Other factors, including ionization mode (i.e., positive or negative) or type of fragmentation (i.e., collision-induced dissociation [CID] or higher-energy C-trap dissociation [HCD]), were also initially considered but were not found to be readily distinguishable from instrument type.

We approached the development of an algorithm to uncover which generative factors correlated with which latent variables in two steps: (1) exhaustive factor pairwise correlation analysis on the selected subset d_i_ of a larger set D, and (2) selected factor pairwise correlation analysis on the complete subset D. The subset D of the dataset is a broader subset, selected by the factors F_i_ with values in the limited range, or list of unique values, e.g., in our analysis, we limited the total exact mass from 200 *m*/*z* to 500 *m*/*z*, the collision energy from 10 to 50, the ionization mode to {positive, negative}, and the instrument type to {1: LC-ESI-QTOF, 0: ESI-QFT, 7: LC-ESI-ITFT, 2: LC-ESI-QFT, 10: Linear Ion Trap}. The subset d_i_ of D is a narrow subset where only one factor F_i_ varies in the limited range at the time (the same range as in the definition of D), and the remaining factors are fixed to a single value (for a discrete factor) or a small interval/step (for a continuous factor). Note that each factor F_i_ has its own narrow subset d_i_, e.g., if F_2_ is the total exact mass, the subset d_2_ is defined by all compounds with the total exact mass between 200 *m*/*z* and 500 *m*/*z*, the collision energy equal to 45, positive ionization mode, and the instrument type set to ESI-QFT. In this analysis, for continuous variables such as the total exact mass and the collision energy, we used a step size of 5 to create discrete bins; together with five different instrument types and two ionization modes, 5490 partitions di of the subset D per each factor were formed.

The first step computes the Pearson correlation coefficient for all pairs of factor F_i_ and latent variable Z_j_ on the subset d_i_. In this analysis, we picked the model that had latent representation with three dimensions; therefore, there were nine cases, i.e., three factors and three latent variables. By letting only one factor vary at a time and fixing other factors to a single value, we controlled for the impact of other factors on the correlation calculation. Figure 9 shows an example of such an analysis. The subset di was selected according to the criteria specified in the bottom table, e.g., for the total exact mass (F_2_), the resulting subset d_2_ had N = 2020 data points from the MoNA dataset. In the first row was collision energy, where we can observe that it correlated best with the latent variable Z_3_. In the next row was the total exact mass, which had the strongest correlation with the latent variable Z_2_. The last factor was the instrument type, which correlated with the latent variable Z_1_. The subplot with the strongest correlation coefficient is highlighted with a yellow background for each factor to indicate which latent variable represented that factor best. It is important that the highest correlations occurred for distinct latent variables, which is depicted as a yellow diagonal in Figure 10. This example is consistent with the assumption regarding disentangled representation [29] and our notion of distinctness, as stated above.

The second step calculates the Pearson correlation coefficient for the complete subset D by accumulating the individual Pearson correlation coefficient for each partition d_i_ by performing the first step for all possible subsets d_i_ associated with the specified subset D. The collective analysis of all cases can be summarized and expressed in terms of distributions of coefficients, which provides an overview of all cases, and determines which factors correlated with latent variables stronger and which correlated weaker. We can say that the factor correlated with the latent variable better on average if the mean of the corresponding distribution was further from 0 and closer to 1 or −1, and the mean was consistent across all partitions di as measured by the standard deviation of this distribution. Similar work has been carried out on computer vision datasets by looking at the correlation between latent dimensions and factors of variation [30].

Figure 11 shows the aggregated results of such analysis for model M2 (Appendix A), where the factors were assigned to the latent variables indicated in the round brackets. We assigned factors to latent variables by the following procedure: first, the factor and latent variable with the highest mean correlation coefficient were matched, e.g., Z_1_ and instrument type from Appendix A. Second, the factor and latent variable with the next highest mean correlation coefficient were matched, e.g., Z_2_ and total exact mass from Appendix A. The matching between factor and latent variables based on the next highest mean correlation coefficient was repeated for all remaining factors and latent variables. Then, for the remaining two factors, we repeated that process, e.g., Z_2_ and total exact mass from Appendix A, and Z_3_ and collision energy from Appendix A. The exact values of Pearson correlation coefficient means are available in Table 2.

The two-step algorithm to uncover correlations between generative factors and latent variables described above demonstrates that an analysis that does not require visual intuition for probing the structure of the latent space for complex and high-dimensional data such as tandem mass spectra can be realized. In addition, the two-step algorithm allows one to identify models that are disentangled with respect to generative factors that the researcher hypothesizes contribute the greatest variation to the input domain and, hence, are more likely to be captured within the latent space. This can be achieved in a data-efficient manner (if we can assume that the data is well distributed in the input space) whereby only subsets of the data need to be probed instead of the complete dataset.

## 4. Conclusions

In this study, a comprehensive investigation of unsupervised disentanglement of tandem mass spectra using variational autoencoders was conducted. An exhaustive exploration of model hyperparameters, data preprocessing parameters, and tandem mass spectra generative factors using classification and regression tasks along with permutation feature importance revealed dependencies between mass spectra generative factors, the importance of particular preprocessing parameters such as the number of peaks, and trade-offs between reconstruction accuracy and disentanglement when modulating the beta and capacity parameters. Importantly, we were able to verify that a sparse vectorized mass and intensity pair was a superior input representation to binning mass spectra according to mass, which causes an inevitable loss in mass resolution, and the latent space vector provides a superior representation of mass spectra compared to the raw input for downstream classification and regression tasks. Qualitative analyses of the latent space using PCA visualization provided evidence of increased compression of the latent space as the KL divergence loss was weighted more compared to the reconstruction loss and provided evidence of unsupervised clustering of generative factors in the latent space. Latent traversals revealed a smooth and continuous shape to the latent space. Latent space interpolations confirmed that the latent space was smooth and semantically meaningful, resulting in plausible interpolant reconstructions. And finally, a semi-exhaustive pairwise correlation analysis method was developed that identified significant and unique correlations between latent variables and generative factors without the need for visual intuition. The qualitative and quantitative exploration of the disentanglement of tandem mass spectra using variational autoencoders demonstrates that unsupervised disentanglement of tandem mass spectra using deep generative modeling is possible and ripe for future innovations and applications.

A major hurdle encountered during this study and in recent studies was the lack of an unsupervised disentanglement metric. We explored the use of supervised metrics commonly used in the literature for simple computer vision tasks and found them to be unreliable for tandem mass spectra. Our developed two-step pairwise correlation analysis is effective when visual intuition cannot be relied upon, as is the case of high-dimensional mass spectral data, but still requires knowledge of the ground truth factors of variation. Our early experiments to understand which generative factors were independent and most important were our attempts to narrow down which factors were most likely leading to the greatest variation in the mass spectra datasets used in this study with minimal bias. Future work will seek to develop a method that is able to both select the most probable causal generative factors and measure how well they correspond to distinct latent dimensions. Recent work on causal representation learning using neural networks [31,32] could be an applicable approach. Our developed two-step pairwise correlation analysis is also data efficient, allowing one to probe a subset of the data instead of the entire dataset. Future work would need to test the limits to the sizes of data subsets to accurately represent the entire dataset.

It is also worth mentioning that this study employed relatively small and simple encoder and decoder architectures compared to those found in state-of-the-art computer vision and natural language processing models. Given the benefits of the sparse vectorized mass and ion intensity pairs on memory efficiency and data expressiveness described in this work, we anticipate that the use of more powerful encoder and decoder architectures will facilitate disentanglement and synthetic mass spectra generation.

## 5. Code Repository

The implementation of the models and the above-mentioned algorithms can be found at https://github.com/ChristopherAbram/SpecVAE (accessed on 30 August 2023). The repository has the following structure: it contains *specvae* Python package with models, criterium classes, trainers, and metrics implementations. The package is used by the training scripts, i.e., entry points with parameters included in the *train* directory. The *train* directory includes training configurations encapsulated in JSON format whose files describe a set of parameters used to run a given train script (experiment). The *notebook* directory holds Jupyter Lab’s notebooks with data analysis, plotting, latent space visualization and traversal, model scores comparison, etc. A detailed description of the functionalities with relevant instructions is enclosed within the repository.

The models are implemented with PyTorch, a deep-learning Python framework. The code can be run either on CPU or GPU, depending on the preferences of the user. Some of the other metrics and algorithms are implemented with standard Python numerical packages, e.g., numpy, scipy, scikit-learn, and similar.

## Figures and Tables

**Figure 1 biomolecules-13-01343-f001:**
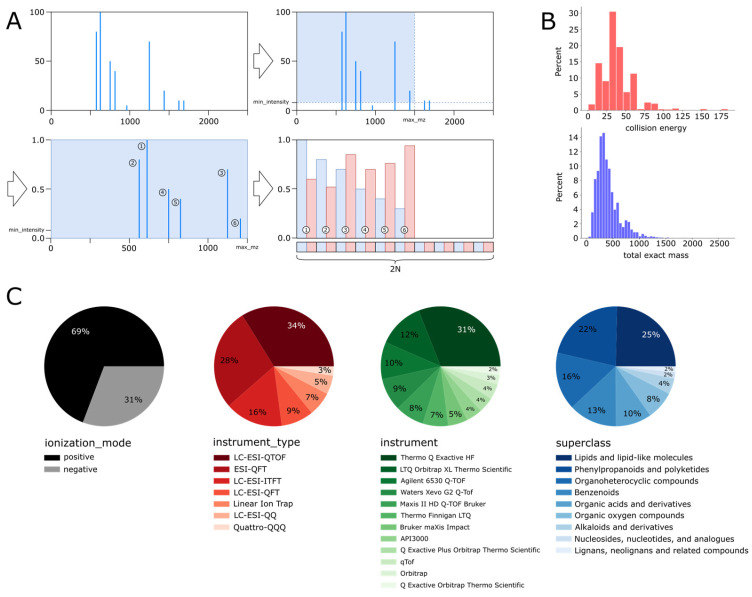
(**A**) Graphical demonstration of the data preprocessing pipeline and spectrum representation. Numbers 1–6 indicate peaks sorted by intensity in descending order. (**B**) Distribution of the collision energy and the total exact mass in the MoNA dataset. (**C**) Class proportions in the MoNA dataset: ionization mode, instrument type, instrument, and superclass, respectively.

**Figure 2 biomolecules-13-01343-f002:**
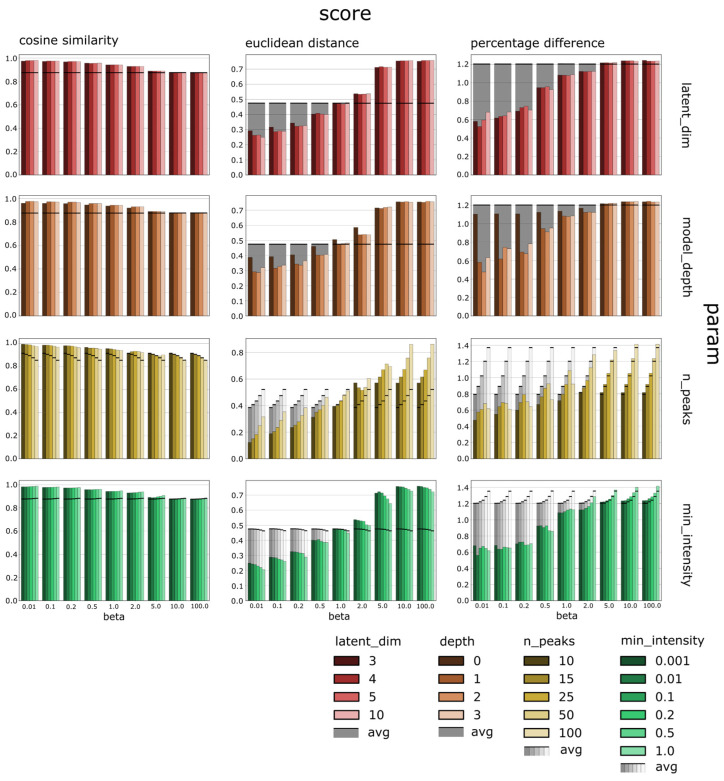
The summary of the reconstruction scores by the metric type (columns), model parameters (two first rows), data preprocessing parameters (two last rows), and regularization strength represented by distinct values of the beta parameter. The shaded bars indicate the corresponding baseline metric average reconstruction score (see Section 2). The reconstruction quality depended mainly on the regularization strength parameter, the maximum number of peaks in the spectra representation, and the size of the latent representation. The latent_dim is the size of the latent space dimension; depth is the model depth and is the maximum number of layers in the encoder and decoder decreased by the number of layers of the backbone architecture which was 2 (e.g., for a symmetric model with 3 linear layers in the encoder and decoder, the depth is 1); n_peaks is the maximum number of peaks in the input spectra; and min_intensity is the minimum normalized intensity of any peak in the input spectra.

**Figure 3 biomolecules-13-01343-f003:**
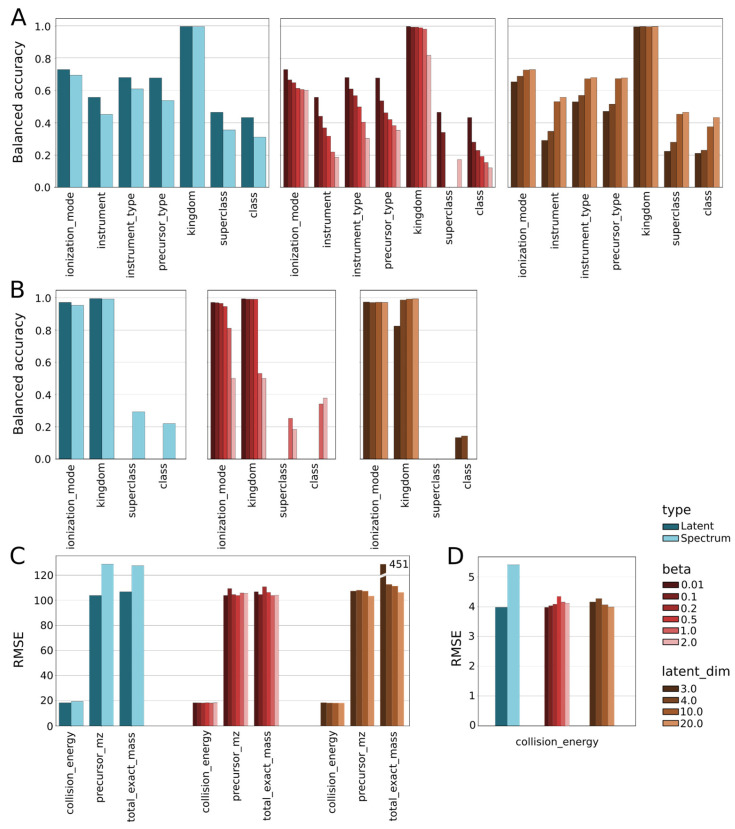
Side-by-side comparison of classification and regression performance in the task predicting MS/MS factors (on the *x*-axis), solely based on the spectrum versus its latent representation. The blue bars show the best-performing model in its category, i.e., latent vs. spectrum; the red bars show the scores by the beta regularization strength; and the orange bars divide metrics by the size of the latent dimension (size of the compression). (**A**) Balanced accuracy in classification tasks in the MoNA dataset. (**B**) Balanced accuracy in classification tasks in the HMDB dataset. (**C**) RMSE values in regression tasks in the MoNA dataset. (**D**) RMSE values in regression tasks in the HMDB dataset.

**Figure 4 biomolecules-13-01343-f004:**
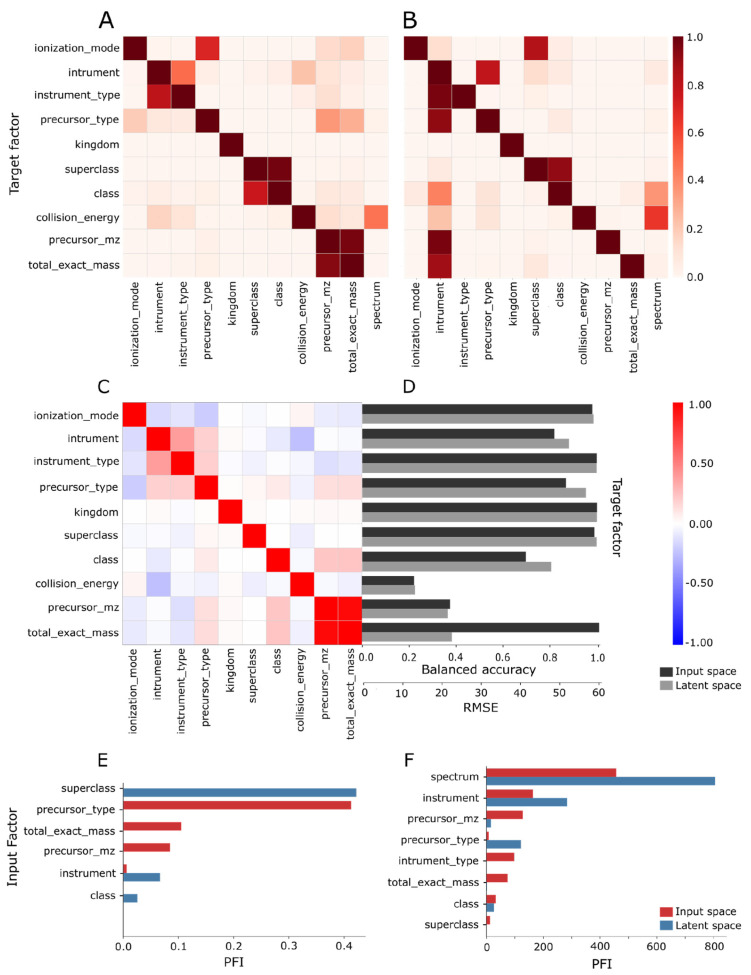
Metrics comparison for classification and regression models trained on the MoNA dataset. (**A**) Heatmap of the relative PFI scores for models trained with spectrum and factors as the input, rows of the matrix designate target variables, and columns represent factors. Additionally, the last column contains PFI scores for the spectrum (see Appendix A). (**B**) Heatmap of the relative PFI scores for models trained jointly with BetaVAE using latent representations, each BetaVAE was configured to use latent_dim = 10 and max_num_peaks = 50 (see Appendix A). (**C**) Pearson’s cross-correlation matrix of the factors in the MoNA database. (**D**) Score comparison between models trained with spectra and models trained on latent representations of spectra. Balanced accuracy scores and RMSE scores for classification and regression models, respectively, for each target factor. (**E**) Comparison of the PFI scores for ionization mode as the target factor between models trained on the input spectra (red) and models trained on the latent representations of spectra (blue). Scores are sorted from the most to the least contributing factor. (**F**) Another comparison of the PFI scores for collision energy as the target factor.

**Figure 5 biomolecules-13-01343-f005:**
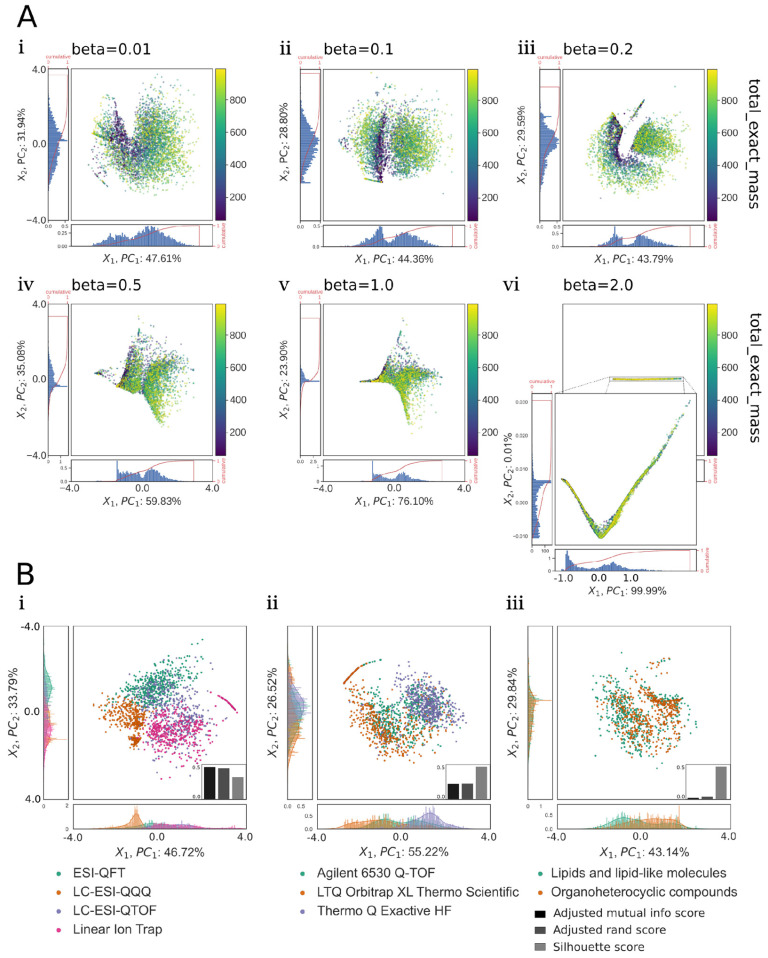
The PCA projections of the 3D latent points encoded by different BetaVAE models from the subset of the MoNA dataset. (**A**) The mosaic of latent space visualizations, colored by the total exact mass, for the model architecture M3 (Appendix A) trained with six different beta values (**i**–**vi**), while other hyperparameters were kept constant. The variance proportion increases for the PC1 dimension as the beta value increases, while the PC2 collapses to nearly zero for beta = 2.0. (**B**) The visualization of the latent space for different BetaVAE models, colored by categorical factors. (**i**) The MoNA dataset clustered by the instrument type in the latent space of the model M6 (Appendix A, beta = 0.01). (**ii**) The MoNA dataset clustered by the instrument vendor in the latent space of the model M6 (Appendix A, beta = 0.01). (**iii**) The MoNA dataset weakly clustered by the compound superclass in the latent space of the model M7 (Appendix A, beta = 0.2).

**Figure 6 biomolecules-13-01343-f006:**
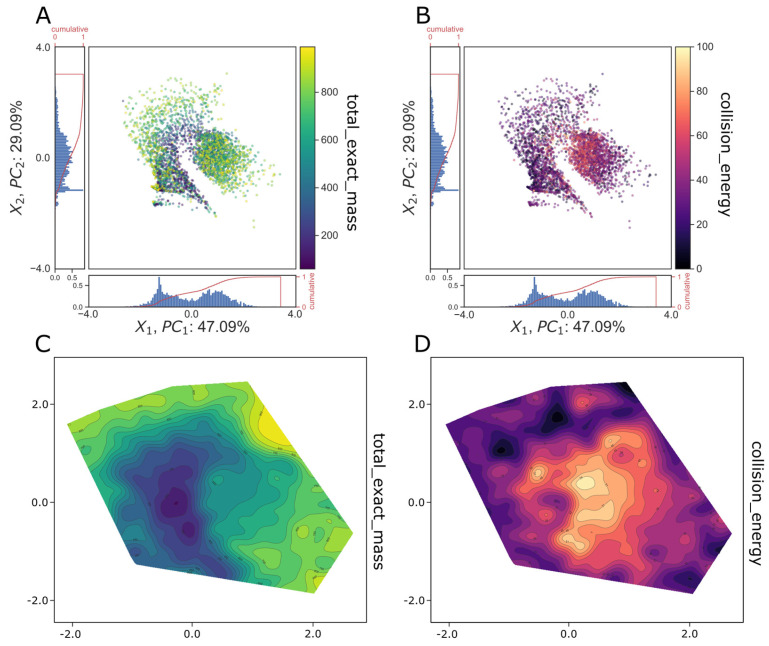
Comparison of the compound distribution in the latent space of the BetaVAE model M4 (Appendix A) colored by two continuous factors: total exact mass and collision energy. The example shows that the latent space was organized such that the density of compounds with high total exact mass and high collision energy were inversely correlated. (**A**) Points in the latent space colored by the total exact mass. (**B**) The same set of points is colored by the collision energy. (**C**) The contour plot for the total exact mass was obtained using triangulation with linear interpolation performed on the original set of latent points. Results were subsequently smoothed by a Gaussian filter to reduce value fluctuations. (**D**) Analogous contour plot for the collision energy.

**Figure 7 biomolecules-13-01343-f007:**
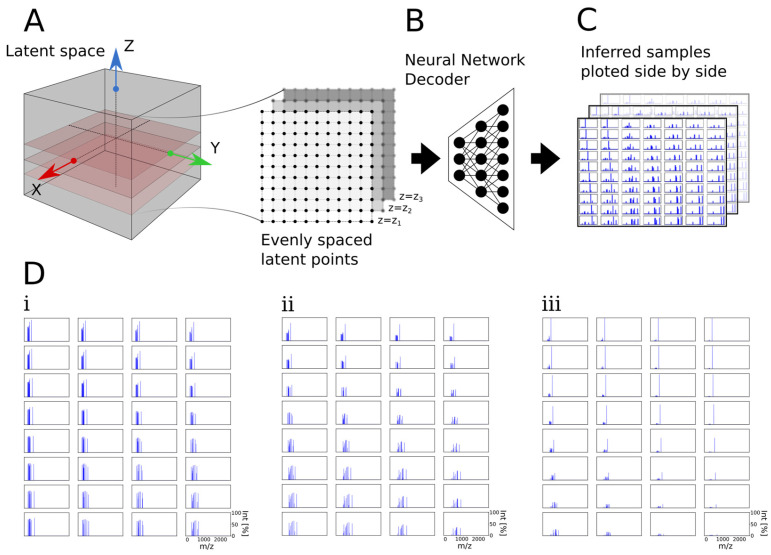
Schematic visualization of the two-step latent traversal method that generated spectra from evenly spaced latent points using the decoder inference network. In this example, points were traversed from evenly spaced latent points in the XY plane by holding the *Z* axis constant. (**A**) Multiple XY plane slices were generated along the *Z* axis centered around the origin. (**B**) Latent points were fed into the decoder network (**C**) to create multiple 2D grids of generated spectra. (**D**) The bottom plots were cropped from different regions (**i**–**iii**) of the larger grid and showed that the adjacent spectra were similar and became more dissimilar as the distance of corresponding latent points increased.

**Figure 8 biomolecules-13-01343-f008:**
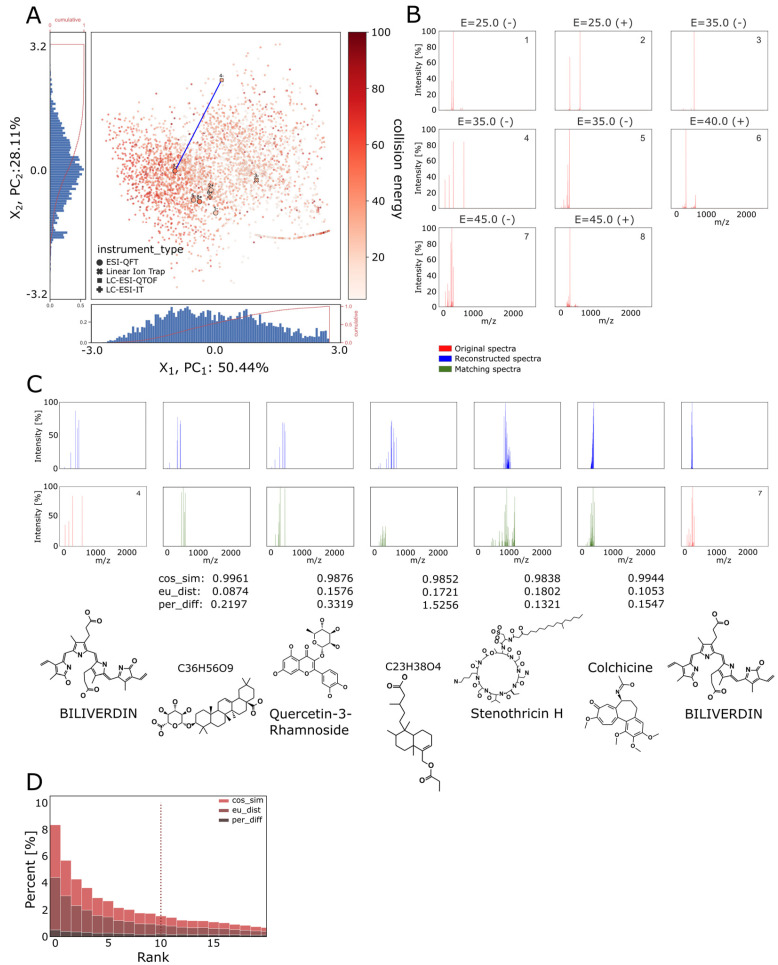
(**A**) Two-dimensional PCA analysis of the BetaVAE (model M6, Appendix A) with a selected compound (Biliverdin) designated with the larger markers. Multiple points for a single molecule represent varying factors, instrument types, and ionization modes. The diagram was rendered using N = 3640 data points and represents the distribution of collision energy in the latent space of the BetaVAE model. (**B**) Spectra of the selected compound, with the collision energy given as the E value, and ionization mode represented as +/− sign. Points can be located in the PCA diagram in panel A by reference numbers. (**C**) slerp interpolation between two selected data points for the same compound and different instrument types, i.e., point no. 4 with LC-ESI-QTOF and point no. 7 with ESI-QFT, symbolically visualized by the bold blue line in panel A. The interpolants are represented with the blue color, with an order from left (point no. 4) to right (point no. 7). The intermediate interpolants are compared against the entire database, and the green spectra match the corresponding interpolants above. The cosine similarity is used as the matching criterion with the associated similarity scores listed below. (**D**) Histogram of the ranked score for the three similarity functions used as the matching criteria, the diagram was generated for N = 20,000 compounds selected from the MoNA dataset without prefiltering based on a chemical formula or exact mass. The rank is computed as the position of the compound in the matching list constructed by comparing the reconstruction of the spectrum for that compound with N other spectra in the database.

**Figure 9 biomolecules-13-01343-f009:**
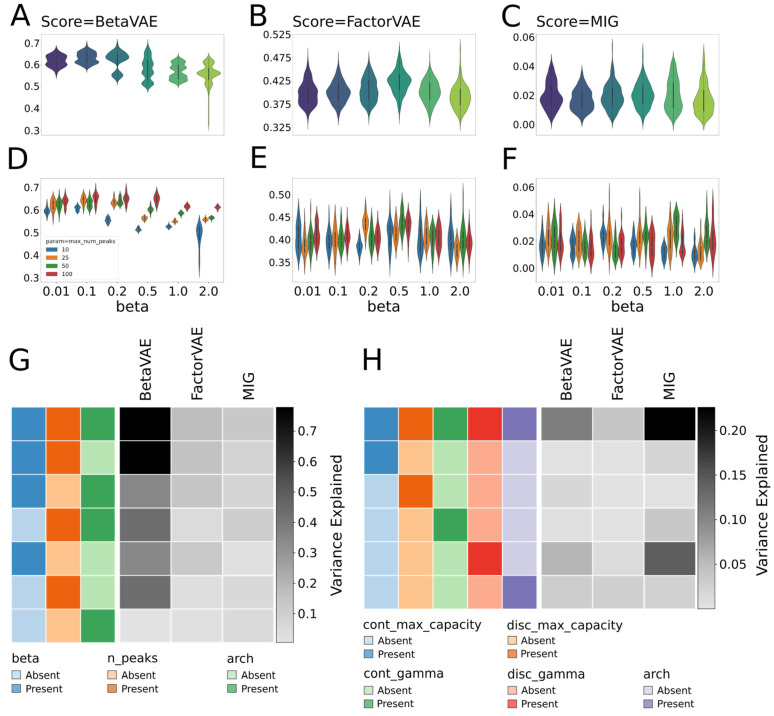
Distributions of disentanglement scores, BetaVAE, FactorVAE, and MIG in columns for different regularization strengths (beta) and the maximum number of peaks (n_peaks or max_num_peaks), evaluated on BetaVAE models on the MoNA dataset. (**A**–**C**) The overall distributions of scores, there is no meaning to the colors; (**D**–**F**) a detailed view of distributions grouped by the max_num_peaks parameter. (**A**,**D**) In the case of the BetaVAE score, we can observe that models trained on the larger number of peaks had higher values of the score, and for the larger values of the beta, i.e., beta *>* 0.2, metrics were less distributed and formed distinct groups (see Results). (**B**,**E**) The FactorVAE scores did not show dependence on beta value, nor were they dependent on the max_num_peaks. (**C**,**F**) Similarly, MIG scores did not show any dependence. (**G**,**H**) Percentage of variance explained regressing the disentanglement scores on different combinations of factors. The results vary depending on which factors were included (present) and which were excluded (absent) from regression. The variance explained metric is computed using the least squares method. (**G**) Variance explained while regressing the disentanglement scores evaluated on the subset of BetaVAE models (Appendix A) on the MoNA dataset using 3 factors: regularization strength: beta; the maximum number of peaks: n_peaks; and/or model architecture: arch. (**H**) Variance explained while regressing the disentanglement scores evaluated on a subset of JointVAE models (Appendix A) on MoNA dataset using 5 factors: maximum continuous capacity: cont_max_capacity; maximum discrete capacity: disc_max_capacity; continuous gamma regularization factor: cont_gamma; discrete gamma regularization factor: disc_gamma; and/or model architecture: arch. JointVAE models were trained on spectra with parameter n_peaks = 50 (the maximum number of peaks).

**Figure 10 biomolecules-13-01343-f010:**
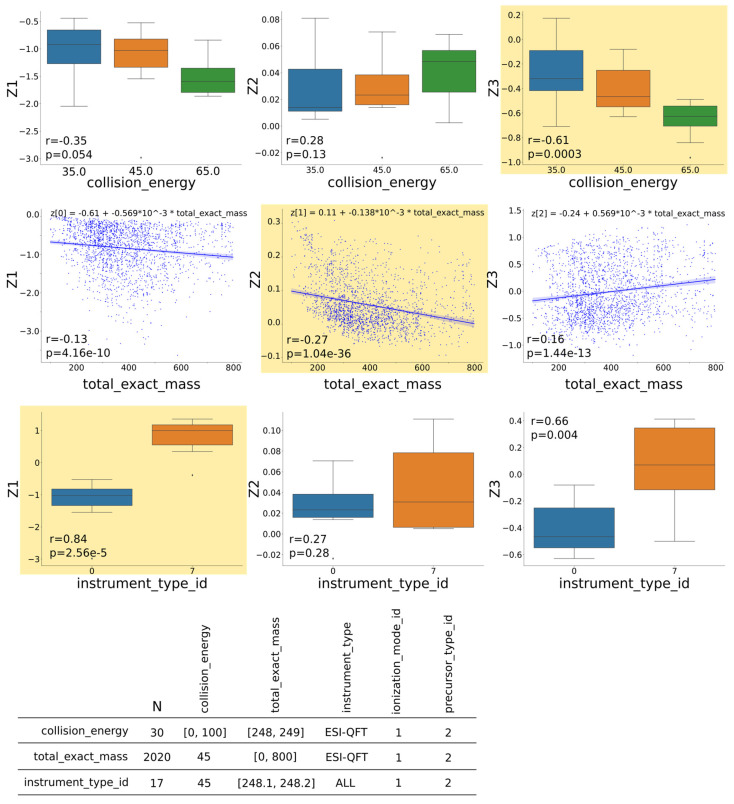
The exhaustive factor pairwise correlation analysis on a small sample of compounds di for points in the 3D latent space of the model M1 (Appendix A). Each latent value (*y*-axis), i.e., Z_1_, Z_2_, and Z_3_, was correlated against each factor (*x*-axis), collision energy, total exact mass, and instrument type, and the Pearson correlation coefficient r and the associated p-value were indicated in the left-bottom corner of each subplot to evaluate the strength of the correlation. Columns of the figure correspond to the single dimension of the latent space, and rows correspond to the factor. The subset of compounds was selected by the criteria specified in the bottom table, and these criteria were different for each factor; the number N specifies the size of each subset. For each factor (row), we highlighted the latent variable (column) that had the strongest correlation with that factor with the yellow color; moreover, we expected it to occur in a distinct column for all factors, like in the figure, assuming that the model attained the disentanglement. Colors are added for better visibility.

**Figure 11 biomolecules-13-01343-f011:**
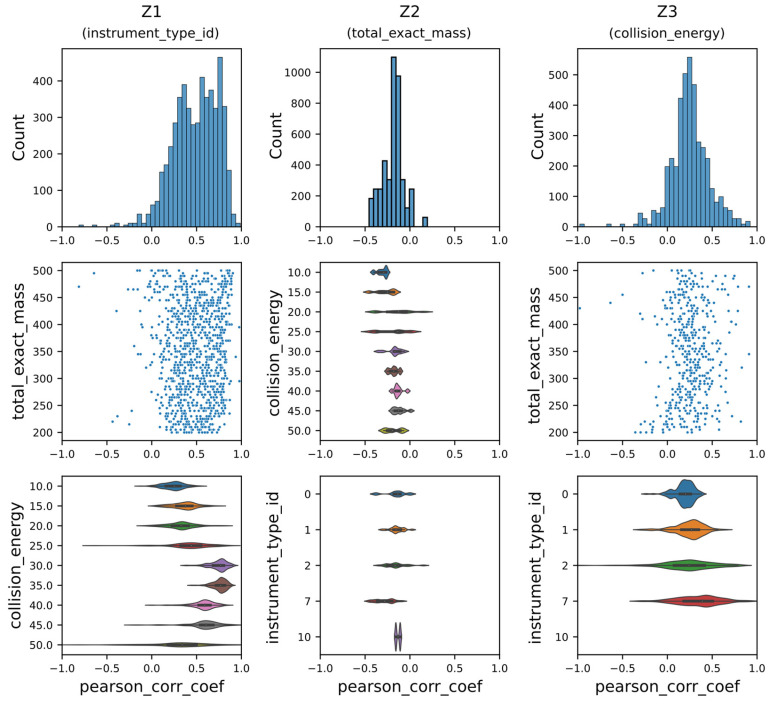
The summary of the correlation analysis of points in the latent space of the model M2 (Appendix A), i.e., select factor pairwise correlation analysis on the complete subset D of the MoNA dataset (as described in the main text and methods). The first column shows the results of the correlation between the first latent value Z_1_ of the latent representation and the instrument type, the second column between Z_2_ and the total exact mass, and the third column between Z_3_ and the collision energy. The first row shows histograms of the Pearson correlation coefficient between respective latent value Z and the associated factor to demonstrate whether the correlation was consistent across subset D or not. The second and the third rows show the same distribution by the values of the remaining two factors, e.g., for the first column, instrument type, we grouped distributions by distinct values of collision energy to observe if the correlation was stronger/weaker for particular groups, or whether it depended on the mass of compounds. Colors are added for better visibility.

**Table 1 biomolecules-13-01343-t001:** The summary of ranked scores shown in Figure 8D.

	Cos_Sim	Eu_Dist	Per_Diff
mean	63.82	63.82	768.95
std.dev	282.86	282.86	1089.56
min	0	0	0
25%	2	2	65
50%	10	10	304
75%	38	38	1064

**Table 2 biomolecules-13-01343-t002:** The average Pearson correlation coefficients from Appendix A. Each factor is assigned to the latent variable Zi, highlighted with bold text.

	Total Exact Mass	Collision Energy	Instrument Type
Z_1_	−0.12	−0.31	**0.49**
Z_2_	**−0.18**	−0.04	−0.02
Z_3_	−0.10	**0.24**	−0.12

## Data Availability

The code and datasets needed to reproduce the analyses in the manuscript can be found at https://github.com/ChristopherAbram/SpecVAE (accessed on 30 August 2023).

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
