# Peer review of "In Search of Disentanglement in Tandem Mass Spectrometry Datasets"

_biomolecules, 2023, doi:10.3390/biom13091343_

Round 1

Reviewer 1 Report

The writers came up with a two-step plan to make it easier to choose untangled models that could be used with other complicated and high-dimensional data sets. They used MS/MS data to do this.

This paper has significant value to data scientists. The paper has writing and design flaws, and below are the major comments on them.

1.    In the paper please defined the abbreviation before its use. E.g. MS/MS, MoNA, HMDB etc. Please write the full meaning of it so that readers can understand it well.

2.    Citation of figures looks very far, example figure 1 has been cited at line number 40, but inserted in line number 409.  See, it’s far away citation. Carefully apply to all tables and figures.

3.    Most important the labeling of paper is very poor. It should be placed under the figures. Please see the structure of dpi template or existing papers.

4.    The Figure caption should be concise and proper. See please Figure 2 has a very long caption. It seems to have an explanation or missed it. This rule applies to all figures.

5.    One more issue I found that the figure and tables are merged together. E.g. figure 8 has a joint of it. Also, table attributes are not defined in the paper such as cos_sim, per_diff.. I suggest explaining these attributes in the paper. This rule also applies to the whole paper either in table or figure.

6.    Create tables and figures separately.  

7.    Some latest references should also included.

Minor language bugs can be rechecked during revision phase. 

Author Response

Comments and Suggestions for Authors

The writers came up with a two-step plan to make it easier to choose untangled models that could be used with other complicated and high-dimensional data sets. They used MS/MS data to do this.

This paper has significant value to data scientists. The paper has writing and design flaws, and below are the major comments on them.

Authors: We would like to express our sincere gratitude for the reviewer’s time and expertise in reviewing the manuscript. The reviewer’s constructive feedback has been invaluable in shaping and improving the quality of our work. We appreciate the thoroughness with which you evaluated the manuscript and provided suggestions for enhancement. Below we provide an elaborated response on how we incorporated the feedback in the revision process for each suggestion.

(1) In the paper please defined the abbreviation before its use. E.g. MS/MS, MoNA, HMDB etc. Please write the full meaning of it so that readers can understand it well.

Authors: We include the full expansion of used abbreviations in the Introduction section, and in-depth definition of the HMDB and MoNA in context of our article in section 2.4.

(2) Citation of figures looks very far, example figure 1 has been cited at line number 40, but inserted in line number 409.  See, it’s far away citation. Carefully apply to all tables and figures.

Authors: We moved figure 1 to line number 44. Citations of figures in the subsection 2.1 are in fact “far-away” from the actual figure placement. Our rational for subsection 2.1 is to provide a concise overview of all experiments for the reader. We chose to also reference the figures in this section for the readers’ convenience. We then present the figures in the results sections where they are discussed.

(3) Most important the labeling of paper is very poor. It should be placed under the figures. Please see the structure of dpi template or existing papers.

Authors: We have addressed this comment to the best of our abilities. We were unsure of what was meant by “labeling”, so we reviewed all of the figure and table element numbers and section labels to ensure they meet the MDPI requirements.

(4) The Figure caption should be concise and proper. See please Figure 2 has a very long caption. It seems to have an explanation or missed it. This rule applies to all figures.

Authors: We appreciate the Reviewer’s concern on the length of the figure captions. We have reviewed the captions for all figures and tables and edited them for conciseness where appropriate. 

(5) One more issue I found that the figure and tables are merged together. E.g. figure 8 has a joint of it. Also, table attributes are not defined in the paper such as cos_sim, per_diff.. I suggest explaining these attributes in the paper. This rule also applies to the whole paper either in table or figure.

Authors: We added explanation of used attributes, i.e., cos_sim, per_diff, in the section 2.3.2 where we first introduce them. We separated the table from figure 8 and placed it in the main content. We decided to keep the table in figure 10 as the table represents the selection criteria for points shown in figure 10 (not overall results), thus it is easier to read the figure.

(6) Create tables and figures separately.

Authors: This issue was fixed in suggestion no. 5.

(7) Some latest references should also included.

Authors: We have added citations on recent tandem mass spectra deep learning methods and investigations of disentanglement where appropriate to the introduction.

Reviewer 2 Report

Ability to use MS/MS spectra with generative modeling in an agnostic digital representation of metabolites would be very important development in the field of rapid identification and classification of molecules using AI based automation. The authors successfully used various auto encoders to untangle data from tandem MS datasets to form a 2-step method for selecting models that are disentangled which can be useful for high-dimensional datasets. Two steps are: (1) exhaustive factor pairwise correlation analysis on the selected subset d(sub i) of a larger set D & (2) selected factor pairwise correlation analysis on the complete subset D.

While it is a type of learning, the two steps are not entirely clear what lead to this thinking. A brief explanation of why this approach over other possibilities would be appreciated. And could a neural network algorithm be useful or considered as an optional way to complete these tasks?

Overall they used a two-step pairwise correlation analysis method for data analysis to expedite and sort their data more efficiently which lead them to conclude that they need to test size limits of this approach in the future to ascertain its accuracy in representation.

Author Response

Comments and Suggestions for Authors

Ability to use MS/MS spectra with generative modeling in an agnostic digital representation of metabolites would be very important development in the field of rapid identification and classification of molecules using AI based automation. The authors successfully used various auto encoders to untangle data from tandem MS datasets to form a 2-step method for selecting models that are disentangled which can be useful for high-dimensional datasets. Two steps are: (1) exhaustive factor pairwise correlation analysis on the selected subset d(sub i) of a larger set D & (2) selected factor pairwise correlation analysis on the complete subset D.

Overall they used a two-step pairwise correlation analysis method for data analysis to expedite and sort their data more efficiently which lead them to conclude that they need to test size limits of this approach in the future to ascertain its accuracy in representation.

Authors: We would like to thank the Reviewer for their time and constructive feedback on the manuscript. We detail our revisions with respect to the specific question about the two step algorithm discussed in the manuscript below.

(1) While it is a type of learning, the two steps are not entirely clear what lead to this thinking. A brief explanation of why this approach over other possibilities would be appreciated. And could a neural network algorithm be useful or considered as an optional way to complete these tasks?

Authors: We added a brief explanation of why we used this specific approach over others, including citations to more specific references exploring similar approach on different datasets (line 941). In our article we do not argue whether implementing our method with neural network would provide any enhancement, however, in the conclusions we strongly encourage readers to pursue further advances of this method, which we provide some guidance about. We would like to point out that we have developed this method keeping in mind its simplicity, i.e., the model which disentangles the dataset is fairly complex, thereby evaluating how well it disentangled the dataset with another deep-based model could substantially decrease the method’s interpretability. In our article, we explore the automatic evaluation of disentanglement with dedicated metrics and we make an argument that using classic visual inspection as means to evaluate disentanglement doesn’t work for data as complex as tandem mass spectra.

Round 2

Reviewer 1 Report

Dear Authors

Thanks for improving the paper.